# `TransTab`: Learning Transferable Tabular Transformers Across Tables

**Zifeng Wang**[1] **and Jimeng Sun**[1,2]

[1] Department of Computer Science, University of Illinois Urbana-Champaign
[2] Carle Illinois College of Medicine, University of Illinois Urbana-Champaign
`{zifengw2,jimeng}@illinois.edu`

## Abstract

Tabular data (or tables) are the most widely used data format in machine learning (ML). However, ML models often assume the table structure keeps fixed in training and testing. Before ML modeling, heavy data cleaning is required to merge disparate tables with different columns. This preprocessing often incurs significant data waste (e.g., removing unmatched columns and samples). How to learn ML models from multiple tables with partially overlapping columns? How to incrementally update ML models as more columns become available over time? Can we leverage model pretraining on multiple distinct tables? How to train an ML model which can predict on an unseen table?

To answer all those questions, we propose to relax fixed table structures by introducing a Transferable Tabular Transformer (`TransTab`) for tables. The goal of `TransTab` is to convert each sample (a row in the table) to a generalizable embedding vector, and then apply stacked transformers for feature encoding. One methodology insight is combining column description and table cells as the raw input to a gated transformer model. The other insight is to introduce supervised and self-supervised pretraining to improve model performance. We compare `TransTab` with multiple baseline methods on diverse benchmark datasets and five oncology clinical trial datasets. Overall, `TransTab` ranks 1.00, 1.00, 1.78 out of 12 methods in supervised learning, feature incremental learning, and transfer learning scenarios, respectively; and the proposed pretraining leads to 2.3% AUC lift on average over the supervised learning.

## 1 Introduction

Tabular data are ubiquitous in healthcare, engineering, advertising, and finance [1, 2, 3, 4]. They are often stored in a relational database as tables or spreadsheets. Table rows represent the data samples, and columns represent the feature variables of diverse data types (e.g., categorical, numerical, binary, and textual). Recent works enhance tabular ML modeling using deep networks [5, 6, 7, 8] or designing self-supervision [2, 9, 10, 11]. Those existing works require the same table structure in training and testing data. However, there can be multiple tables sharing partially overlapped columns in the real world. Hence, learning across tables is inapplicable. The traditional remedy is to perform data cleaning by removing non-overlapping columns and mismatched samples before training any ML models, which waste data resources [12, 13, 14]. Therefore, learning across tables with disparate columns and transferring knowledge across tables are crucial to extending the success of deep learning/pretraining to the tabular domain.

Tables are highly structured yet flexible. The first step to achieve learning across tables is to rethink the *basic elements* in tabular data modeling. In computer vision, the basic elements are pixels [15] or patches, [16, 17]; in natural language processing (NLP), the basic elements are words [18] or

36th Conference on Neural Information Processing Systems (NeurIPS 2022).

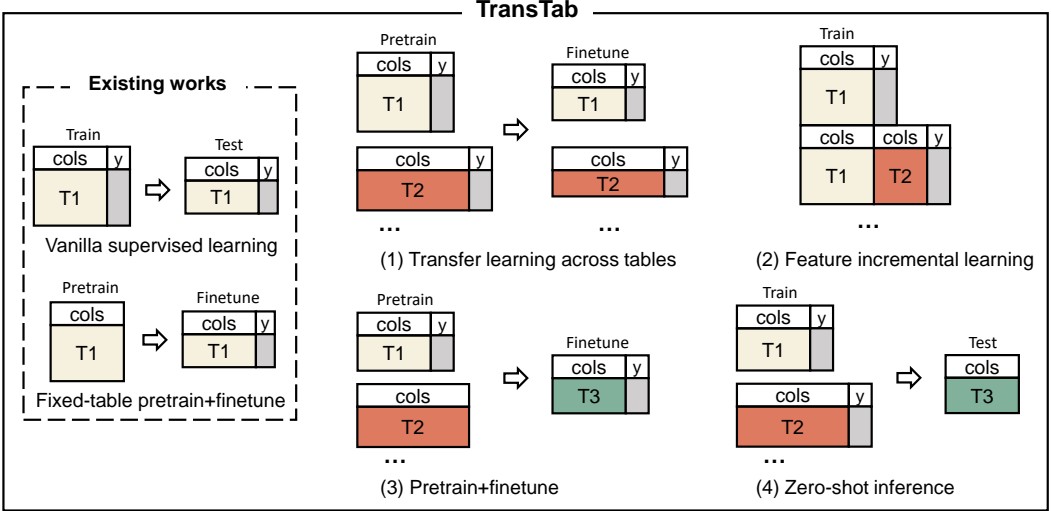

Figure 1: The demonstration of ML modeling on different tabular data settings. Previous tabular methods only do vanilla supervised training or pretraining on the same table due to they only accept **fixed-column tables**. By contrast, `TransTab` covers more new tasks (1) to (4) as it accepts **variable-column** tables. Details are presented in §2.1.

tokens [19, 20]. In the tabular domain, it is natural to treat cells in each column as independent elements. Columns are mapped to unique indexes then models take the cell values for training and inference. The premise of this modeling formulation is to keep the same column structure in all the tables. But tables often have divergent protocols where the nomenclatures of columns and cells differ. By contrast, our proposed work contextualizes the columns and cells. For example, previous methods represent a cell valued *man* under the column *gender* by 0 referring to the codebook {man : 0, woman : 1}. Our model converts the tabular input into a sequence input (e.g., *gender is man*), which can be modeled with downstream sequence models. We argue such featurizing protocol is generalizable across tables, thus enabling models to apply to different tables.

In a nutshell, we propose **Trans**ferable **Trans**formers for **Tab**ular analysis (`TransTab`), a versatile tabular learning framework [1]. `TransTab` applies to multiple use cases as shown in Fig. 1. The key contributions behind `TransTab` are

- A systematic featurizing pipeline considering both column and cell semantics which is shared as the fundamental protocol across tables.
- **V**ertical-**P**artition **C**ontrastive **L**earning (VPCL) that enables pretraining on multiple tables and also allows finetuning on target datasets.

As shown by Fig. 1, due to the fixed-column assumption, all existing works only handle supervised learning or pretraining on the same-structure tables. On the contrary, `TransTab` relaxes this assumption and applies to four additional scenarios, which we will elaborate on in §2.1.

## 2 Method

In this section, we present the details of `TransTab`. Fig. 2 illustrates its workflow including the following key components: 1) The *input processor* featurizes and embeds arbitrary tabular inputs to token-level embeddings; 2) The stacked gated transformer layers further encode the token-level embeddings; 3) Finally, the *learning* module includes a *classifier* trained on labeled data and a *projector* for contrastive learning. Next we will present the details of each component.

---

[1]Our package is available at `https://github.com/RyanWangZf/transtab` with documentation at `https://transtab.readthedocs.io/en/latest/`.

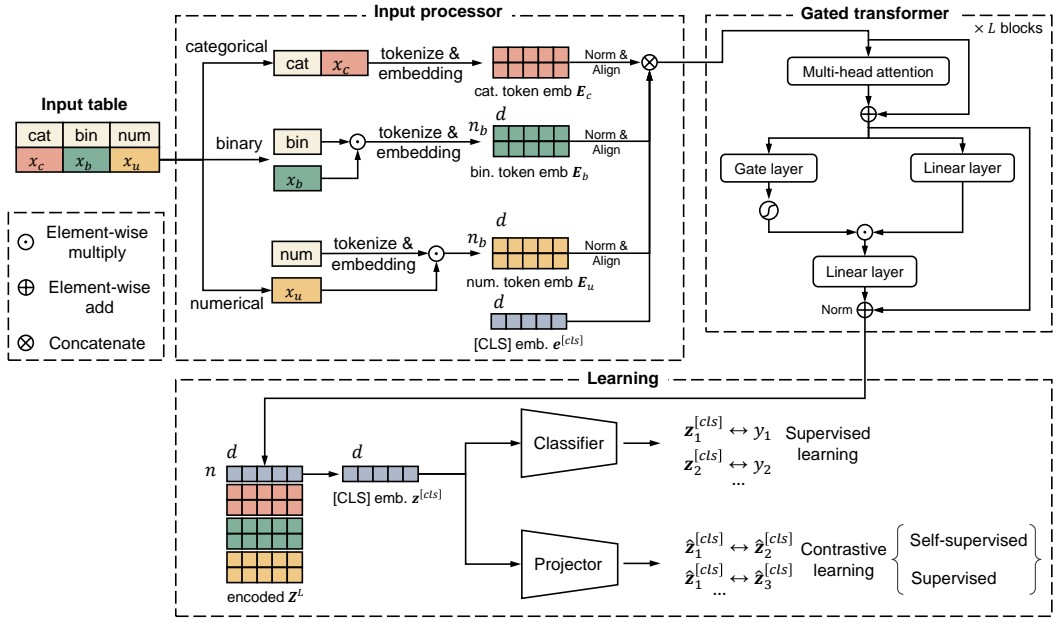

Figure 2: The demonstration of `TransTab` framework. the *input processor* encodes the sample into the token-level embedding $\mathbf{E}$; the `[cls]` embedding $\mathbf{z}^{[cls]}$ in the representation $\mathbf{Z}^L$ after $L$ *gated transformer* layers is used for prediction and learning. In supervised learning, $\mathbf{z}^{[cls]}$ is leveraged by a classifier to make predictions of target $y$; in contrastive learning, the projected $\hat{\mathbf{z}}^{[cls]}$ is is used for self or supervised contrastive loss.

## 2.1 Application scenarios of `TransTab`

Before presenting our method in details, we first introduce four novel applications scenarios which are tractable by `TransTab`, as shown in Fig. 1. Suppose we aim to predict the treatment efficacy for breast cancer trials using multiple clinical trial tables, here are several scenarios we often encounter.

**S(1) Transfer learning.** We collect data tables from multiple cancer trials for testing the efficacy of the same drug on different patients. These tables were designed independently with overlapping columns. How do we learn ML models for one trial by leveraging tables from all trials?

**S(2) Incremental learning.** Additional columns might be added over time. For example, additional features are collected across different trial phases. How do we update the ML models using tables from all trial phases?

**S(3) Pretraining+Finetuning.** The trial outcome label (e.g., mortality) might not be always available from all table sources. Can we benefit pretraining on those tables without labels? How do we finetune the model on the target table with labels?

**S(4) Zero-shot inference.** We model the drug efficacy based on our trial records. The next step is to conduct inference with the model to find patients that can benefit from the drug. However, patient tables do not share the same columns as trial tables so direct inference is not possible.

Overall, we witness that the assumption of fixed table structure is the obstacle to use ML for various applications. Next we will present `TransTab` and demonstrate how it addresses these scenarios.

## 2.2 Input processor for columns and cells

We build the input processor (1) to accept variable-column tables (2) to retain knowledge across tabular datasets. The idea is to convert tabular data (cells in columns) into a sequence of semantically encoded tokens. We utilize the following observation to create the sequence: the column description (e.g., column name) decides the meaning of cells in that column. For example, if a cell in column *smoking history* has value 1, it indicates the individual has a smoking history. Similarly, cell value 60 in column *weight* indicates 60 kg in weight instead of 60 years old. Motivated by the discussion, we

propose to include column names into the tabular modeling. As a result, `TransTab` treats any tabular data as the composition of three elements: text (for categorical & textual cells and column names), continuous values (for numerical cells), and boolean values (for binary cells) . Fig. 2 illustrates a visual example of how these elements are leveraged to process the four basic types of features: categorical/textual `cat`, binary `bin`, and numerical `num`.

**Categorical/Textual feature.** A category or textual feature contains a sequence of text tokens. For the categorical feature `cat`, we concatenate the column name with the feature value $x_c$, which forms as a sequence of tokens. This sentence is then tokenized and matched to the token embedding matrix to generate the feature embedding $\mathbf{E}_c \in \mathbb{R}^{n_c \times d}$ where $d$ is the embedding dimension and $n_c$ is the number of tokens.

**Binary feature.** The binary feature `bin` is usually an assertive description and its value $x_b \in \{0, 1\}$. If $x_b = 1$, then `bin` is tokenized and encoded to the embeddings $\mathbf{E}_b \in \mathbb{R}^{n_b \times d}$; if not, it will not be processed to the subsequent steps. This design significantly reduces the computational and memory cost when the inputs have high-dimensional and sparse one-hot features.

**Numerical feature.** We do not concatenate column names and values for numerical feature because the tokenization-embedding paradigm was notoriously known to be bad at discriminating numbers [21]. Instead, we process them separately. `num` is encoded as same as `cat` and `bin` to get $\mathbf{E}_{u,col} \in \mathbb{R}^{n_u \times d}$. We then multiply the numerical features with the column embedding to yield the numerical embedding as $\mathbf{E}_u = x_u \times \mathbf{E}_{u,col}$[2], which we identify gets an edge on more complicated numerical embedding techniques empirically.

At last, $\mathbf{E}_c, \mathbf{E}_u, \mathbf{E}_b$ all pass the layer normalization [22] and the same linear layer to be aligned to the same space, then are concatenated with `[cls]` embedding to yield $\mathbf{E} = \tilde{\mathbf{E}}_c \otimes \tilde{\mathbf{E}}_u \otimes \tilde{\mathbf{E}}_b \otimes \mathbf{e}^{[cls]}$.

As a result, all cell values are contextualized regarding the corresponding column properties thus the semantic meaning of one element can vary depending on the context composition. This formulation benefits the knowledge transfer across tables a lot. For example, *previously smoked* depicts the same thing as *smoking history*. Previous methods never capture this connection while it is possible for `TransTab` to learn to recognize that $1$ under both columns are equivalent.

## 2.3 Gated transformers

The gated tabular transformer is an adaption of the classical transformer in NLP [23]. It consists of two main components: multi-head self-attention layer and gated feedforward layers. The input representation $\mathbf{Z}^l$ at the $l$-th layer is first adopted for exploring interactions between features:

$$\mathbf{Z}_{att}^l = \texttt{MultiHeadAttn}(\mathbf{Z}^l) = [\text{head}_1, \text{head}_2, \ldots, \text{head}_h]\mathbf{W}^O, \tag{1}$$

$$\text{head}_i = \texttt{Attention}(\mathbf{Z}^l \mathbf{W}_i^Q, \mathbf{Z}^l \mathbf{W}_i^K, \mathbf{Z}^l \mathbf{W}_i^V), \tag{2}$$

where $\mathbf{Z}^0 = \mathbf{E}$ at the first layer; $\mathbf{W}^O \in \mathbb{R}^{d \times d}$; $\{\mathbf{W}_i^Q, \mathbf{W}_i^K, \mathbf{W}_i^V\}$ are weight matrices (in $\mathbb{R}^{d \times \frac{d}{h}}$) of query, key, value of the $i$-th head self-attention module.

The multi-head attention output $\mathbf{Z}_{att}^l$ is further transformed by a token-wise gating layer as $\mathbf{g}^l = \sigma(\mathbf{Z}_{att}^l \mathbf{w}^G)$, where $\sigma(\cdot)$ is a sigmoid function; $\mathbf{g}^l \in [0,1]^n$ controls the magnitude of each token embedding before $\mathbf{Z}_{att}$ goes to the linear projection. This gates then filters the linear layer output

$$\mathbf{Z}^{l+1} = \texttt{Linear}\left((\mathbf{g}^l \odot \mathbf{Z}_{att}^l) \oplus \texttt{Linear}(\mathbf{Z}_{att}^l)\right) \tag{3}$$

to obtain the transformer output $\mathbf{Z}^{l+1} \in \mathbb{R}^{n \times d}$. This mechanism is learnt to focus on important features by redistributing the attention on tokens. The final `[cls]` embedding $\mathbf{z}^{[cls]}$ at the $L$-th layer is used by the classifier for prediction.

## 2.4 Self-supervised and supervised pretraining of `TransTab`

The input processor accepts variable-column tables, which opens the door for tabular pretraining on heterogeneous tables. In detail, `TransTab` is feasible for *self-supervised* and *supervised pretraining*.

**Self-supervised** VPCL. Most SSL tabular methods work on the whole fixed set of columns [2, 24, 11], which take high computational costs and are prone to overfitting. Instead, we take tabular vertical

---

[2]$x_u$ is standardized or normalized in preprocessing.

partitions to build positive and negative samples for CL under the hypothesis that the powerful representation should model view-invariant factors. In detail, we subset columns as illustrated by Fig. 3 where Self-VPCL is on the top right. Suppose a sample $\mathbf{x}_i = \{\mathbf{v}_i^1, \ldots, \mathbf{v}_i^K\}$ with $K$ partitions $\mathbf{v}_i^k$. Neighbouring partitions can have overlapping regions which are justified by the percentage of columns of the partition. Self-VPCL takes partitions from the same sample as the positive and others as the negative:

$$\ell(\mathbf{X}) = -\sum_{i=1}^{B}\sum_{k=1}^{K}\sum_{k'\neq k}^{K} \log \frac{\exp\psi(\mathbf{v}_i^k, \mathbf{v}_i^{k'})}{\sum_{j=1}^{B}\sum_{k^\dagger=1}^{K}\exp\psi(\mathbf{v}_i^k, \mathbf{v}_j^{k^\dagger})}, \tag{4}$$

where $B$ is the batch size; $\psi(\cdot, \cdot)$ is the cosine similarity function. $\psi$ applies to $\hat{\mathbf{z}}^{[cls]}$ which is the linear projection of partition $\mathbf{v}$'s embedding $\mathbf{z}^{[cls]}$. Compared with vanilla CL like SCARF [11], Self-VPCL significantly expand the positive and negative sampling for learning more robust and rich embeddings. What is more, this vertical partition sampling is extremely friendly to column-oriented databases [25] which support the fast querying a subset of columns from giant data warehouses. For the sake of computational efficiency, when $K > 2$, we randomly sample two partitions.

**Supervised** VPCL. When we own labeled tabular data for pretraining, one natural idea would be taking task-specific predicting heads for pretraining on vanilla supervised loss, e.g., cross-entropy loss. In finetuning, these heads are dropped and a new head will be added on top of the pretrained encoder. However, we argue it is suboptimal and may undermine the model transferability. The reason behind is that tabular datasets vary dramatically in size, task definition, and class distributions. Pretraining TransTab using supervised loss inevitably causes the encoder biased to the major tasks and classes. Moreover, the suitable hyperparameter range is often distinct across tabular data when applying supervised loss. The same set of hyperparameters can cause overfitting on one dataset and underfitting on another. Therefore, it is tricky to pick appropriate hyperparameters for pretraining based on vanilla supervised loss.

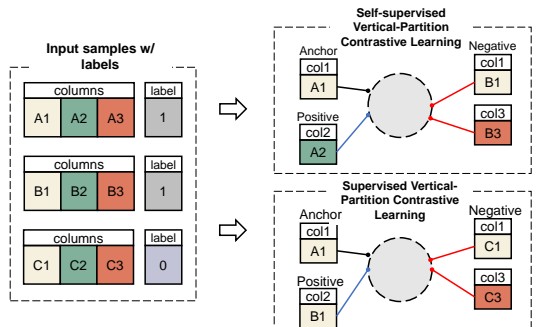

Figure 3: The demonstration of contrastive learning methods (different pieces can either be distinct or be overlapped partially). Self-VPCL: Positive pairs are partitions of the same sample; VPCL: Positive pairs are partitions of the sample belonging to the same class.

In this paper, we propose VPCL for pretraining inspired by supervised CL [26] which was proved robust to noise and hyperparameters. As illustrated by Fig. 3, we build positive pairs considering views from the same class except for only from the same sample:

$$\ell(\mathbf{X}, \mathbf{y}) = -\sum_{i=1}^{B}\sum_{j=1}^{B}\sum_{k=1}^{K}\sum_{k'=1}^{K} \mathbf{1}\{y_j = y_i\} \log \frac{\exp\psi(\mathbf{v}_i^k, \mathbf{v}_j^{k'})}{\sum_{j^\dagger=1}^{B}\sum_{k^\dagger=1}^{K}\mathbf{1}\{y_{j^\dagger}\neq y_i\}\exp\psi(\mathbf{v}_i^k, \mathbf{v}_{j^\dagger}^{k^\dagger})}. \tag{5}$$

$\mathbf{y} = \{y_i\}_i^B$ are labels; $\mathbf{1}\{\cdot\}$ is indicator function. VPCL relieves multiple pretraining predictors required to adjust to different datasets. Moreover, VPCL exposes more feature embeddings to the supervision by partitioning hence providing more discriminative and generalizable representations.

## 3 Experiments

In this section, we aim at answering the following questions by extensive experiments:

- **Q1.** How does TransTab perform compared with baselines under the vanilla supervised setting?
- **Q2.** How well does TransTab address incremental columns from a stream of data (S(2) in Fig. 1)?
- **Q3.** How is the impact of TransTab learned from multiple tables (with different columns) drawn from the same domain on its predictive ability (S(1) in Fig. 1)?

Table 1: Statistics of the use clinical trial mortality prediction datasets. All are binary classification tasks. Positive ratio means the ratio of data points belong the positive class. NCTxxx are trial identifiers which can be linked to trials on ClinicalTrials.gov.

| Name | Datapoints | Categorical | Binary | Numerical | Positive ratio |
|------|-----------|-------------|--------|-----------|----------------|
| NCT00041119 | 3871 | 5 | 8 | 2 | 0.07 |
| NCT00174655 | 994 | 3 | 31 | 15 | 0.02 |
| NCT00312208 | 1651 | 5 | 12 | 6 | 0.19 |
| NCT00079274 | 2968 | 5 | 8 | 3 | 0.12 |
| NCT00694382 | 1604 | 1 | 29 | 11 | 0.45 |

- **Q4.** Can `TransTab` be a zero-shot learner when pretrained on tables and infer on a new table (S(4) in Fig. 1)?
- **Q5.** Is the proposed vertical partition CL better than vanilla supervised pretraining and self-supervised CL (S(3) in Fig. 1)?

**Datasets.** We introduce *clinical trial mortality prediction datasets* where each includes a distinct group of patients and columns [3]. The data statistics are in Table 1. Accurately predicting the patient mortality in clinical trials is crucial because it helps identify catastrophic treatment then save patients from harm and improve the clinical trial design. Considering they are from a similar domain, we can utilize them to test if `TransTab` can achieve transfer learning. Besides, we also include a set of public tabular datasets, the statistics are in Table 7.

**Dataset pre-processing.** For all baselines, we represent categorical features by ordinal encoding if they need to specify categorical features, otherwise one-hot encoding is used. Numerical features are scaled to $[0, 1]$ by min-max normalization. Exceptionally for `TransTab`, we map the categorical feature index to its original description, e.g., mapping class "1" under "gender" to "female".

**Model and implementation protocols.** Unless specified otherwise, we keep the settings fixed across all experiments. `TransTab` uses 2 layers of gated transformers where the embedding dimensions of numbers and tokens are 128, and the hidden dimension of intermediate dense layers is 256. The attention module has 8 heads. We choose ReLU activations and do not activate dropout. We train `TransTab` using Adam optimizer [27] with learning rate in $\{2e\text{-}5, 5e\text{-}5, 1e\text{-}4\}$ and no weight decay; batch size is in $\{16, 64, 128\}$. We set a maximum self-supervised pretraining epochs of 50 and supervised training epochs of 100. A patience of 10 is kept for supervised training for early stopping. Experiments were conducted with one RTX3070 GPU, i7-10700 CPU, and 16GB RAM.

**Baselines.** We include the following baselines for comparison: *Logistic regression (LR)*; *XGBoost* [28]; *Multilayer perceptron (MLP)*; *SeLU MLP (SNN)* [29]; *TabNet* [30]; *DCN* [1]; *AutoInt* [31]; *Tab-Transformer* [5]; *FT-Transformer* [32]; *VIME* [2]; *SCARF* [11]. We provide the baseline architectures and implementations in Appendix B.

## 3.1 Q1. Supervised learning

Results of supervised learning on clinical trial mortality prediction datasets are summarized by Table 2. Note that all methods including ours do not perform pre-training. We see that our method outperforms baselines on all. From the view of method ranks, we surprisingly identify that LR wins over half of baseline methods. Except for `TransTab`, FT-Transformer is the only model that shows significant superiority over LR, which illustrates the potential of transformers for tabular modeling. Additional results on public datasets are available in Table 8 where we witness that our method is comparable to the state-of-the-art baseline tabular models. We also discover the baselines drawn from the CTR prediction literature (DCN and AutoInt) turn out the be competitive in tabular modeling.

## 3.2 Q2. Feature incremental learning

For previous tabular models, we should either drop new features or drop old data when confronting feature incremental learning. By contrast, `TransTab` is able to continually learn from new data with

---

[3]https://data.projectdatasphere.org/projectdatasphere/html/access

Table 2: Test AUROC results on clinical trial mortality datasets the under **supervised learning** setting. All the remaining tables in this paper follow these setups to avoid clutter: the metric values are averaged over 10 random seeds; the *Rank* column reports the average rank across all datasets; Top results for each dataset are in bold.

| Methods | N00041119 | N00174655 | N00312208 | N00079274 | N00694382 | Rank(Std) |
|---------|-----------|-----------|-----------|-----------|-----------|-----------|
| LR | 0.6364 | 0.8543 | 0.7382 | 0.7067 | 0.7360 | 5.40(1.14) |
| XGBoost | 0.5937 | 0.5000 | 0.6911 | 0.6784 | 0.7440 | 9.60(3.71) |
| MLP | 0.6340 | 0.6189 | 0.7427 | 0.6967 | 0.7063 | 8.00(2.83) |
| SNN | 0.6335 | 0.9130 | 0.7469 | 0.6948 | 0.7246 | 5.80(2.39) |
| TabNet | 0.5856 | 0.5401 | 0.6910 | 0.6031 | 0.7113 | 11.40(0.89) |
| DCN | 0.6349 | 0.7577 | 0.7431 | 0.6952 | 0.7458 | 5.60(2.51) |
| AutoInt | 0.6327 | 0.7502 | 0.7479 | 0.6958 | 0.7411 | 6.20(2.59) |
| TabTrans | 0.6187 | 0.9035 | 0.7069 | 0.7178 | 0.7229 | 7.20(3.56) |
| FT-Trans | 0.6372 | 0.9073 | 0.7586 | 0.7090 | 0.7231 | 4.20(2.28) |
| VIME | 0.6397 | 0.8533 | 0.7227 | 0.6790 | 0.7232 | 7.00(3.08) |
| SCARF | 0.6248 | 0.9310 | 0.7267 | 0.7176 | 0.7103 | 6.60(3.91) |
| TransTab | **0.6408** | **0.9428** | **0.7770** | **0.7281** | **0.7648** | **1.00(0.00)** |

Table 3: Test AUROC results on clinical trial datasets under **feature incremental learning**.

| Methods | N00041119 | N00174655 | N00312208 | N00079274 | N00694382 | Rank(Std) |
|---------|-----------|-----------|-----------|-----------|-----------|-----------|
| LR | 0.6213 | 0.8485 | 0.6801 | 0.6258 | 0.7236 | 4.6(3.21) |
| XGBoost | 0.5735 | 0.7890 | 0.6760 | 0.6038 | 0.6463 | 8.8(2.59) |
| MLP | 0.6371 | 0.7754 | 0.6871 | 0.6220 | 0.6851 | 6.2(2.95) |
| SNN | 0.5765 | 0.7440 | 0.6854 | 0.6336 | 0.7035 | 6.4(2.30) |
| TabNet | 0.5548 | 0.8419 | 0.5849 | 0.6052 | 0.6668 | 9.0(3.39) |
| DCN | 0.5172 | 0.5846 | 0.6640 | 0.6535 | 0.6957 | 8.2(4.16) |
| AutoInt | 0.5232 | 0.6075 | 0.7031 | 0.6394 | 0.6974 | 7.2(3.56) |
| TabTrans | 0.5599 | 0.7652 | 0.6433 | 0.6365 | 0.6841 | 8.2(1.10) |
| FT-Trans | 0.5552 | 0.8045 | 0.7148 | 0.6471 | 0.6815 | 5.8(3.11) |
| VIME | 0.6101 | 0.8114 | 0.3705 | 0.6444 | 0.6436 | 7.4(4.22) |
| SCARF | 0.5996 | 0.6261 | 0.7072 | 0.6535 | 0.6957 | 5.2(2.97) |
| TransTab | **0.6797** | **0.8545** | **0.7617** | **0.6857** | **0.7795** | **1.0(0.00)** |

incremental features. We split the raw dataset into three subsets: set1, 2, and 3 which mimic the incremental feature scenario shown by (2) in Fig. 1. Baseline methods apply to two scenarios: (a) learning from all data that only have features of set1 and (b) learning from the data of set3 only. We report the best of the two. TransTab applies to learning from all three subsets. Table 3 shows the results where we find our method outperforms baselines by a great margin. It demonstrates that TransTab makes the best of incremental features to learn better. Similar observations appear in public datasets, shown by Table 9.

## 3.3 Q3. Transfer learning

We further test if TransTab is able to transfer knowledge across tables. Results are in Table 4. We split each dataset into two subsets with 50% overlaps of their columns. Baselines are trained and tested on set1 (only label-supervision) or set2 separately. For our method we pretrain it on set1 then finetune it on set2 and report its performance on set2, and vice versa. We observe that TransTab can benefit from knowledge transfer across tables to reach superior performances. Similar observations are made on public datasets shown by Table 10.

Table 4: Test AUROC results on clinical trial datasets under **transfer learning** across tables.

| Methods | N00041119 | | N00174655 | | N00312208 | | N00079274 | | N00694382 | | Rank(Std) |
|---|---|---|---|---|---|---|---|---|---|---|---|
| | set1 | set2 | set1 | set2 | set1 | set2 | set1 | set2 | set1 | set2 | |
| LR | 0.625 | 0.647 | 0.789 | 0.819 | 0.701 | 0.735 | 0.635 | 0.685 | 0.675 | 0.763 | 5.33(1.73) |
| XGBoost | 0.638 | 0.575 | 0.574 | **0.886** | 0.690 | 0.700 | 0.596 | 0.647 | 0.592 | 0.677 | 7.56(3.75) |
| MLP | 0.639 | 0.621 | 0.314 | 0.857 | 0.683 | 0.744 | 0.620 | 0.675 | 0.648 | 0.765 | 6.56(3.32) |
| SNN | 0.627 | 0.634 | 0.215 | 0.754 | 0.687 | 0.732 | 0.631 | 0.683 | 0.651 | 0.759 | 7.44(2.07) |
| TabNet | 0.564 | 0.558 | 0.856 | 0.592 | 0.671 | 0.657 | 0.443 | 0.605 | 0.581 | 0.677 | 10.67(2.96) |
| DCN | 0.636 | 0.625 | 0.767 | 0.790 | 0.711 | 0.698 | 0.682 | 0.664 | 0.658 | 0.737 | 6.33(2.45) |
| AutoInt | 0.629 | 0.630 | 0.843 | 0.730 | 0.725 | 0.698 | 0.679 | 0.665 | 0.686 | 0.661 | 5.89(2.89) |
| TabTrans | 0.616 | 0.647 | 0.866 | 0.822 | 0.675 | 0.677 | 0.618 | 0.702 | 0.652 | 0.718 | 6.22(3.38) |
| FT-Trans | 0.627 | 0.641 | 0.836 | 0.858 | 0.720 | 0.741 | **0.692** | 0.692 | 0.652 | 0.740 | 4.22(2.28) |
| VIME | 0.603 | 0.625 | 0.312 | 0.726 | 0.601 | 0.642 | 0.477 | 0.668 | 0.614 | 0.715 | 10.44(1.51) |
| SCARF | 0.635 | **0.657** | 0.651 | 0.814 | 0.653 | 0.686 | 0.682 | 0.701 | 0.671 | **0.776** | 5.56(3.40) |
| TransTab | **0.653** | 0.653 | **0.904** | 0.846 | **0.730** | **0.756** | 0.680 | **0.711** | **0.747** | 0.774 | **1.78(1.30)** |

Table 5: Test AUROC results on clinical trial datasets under **zero-shot learning** setting.

| TransTab | N00041119 | N00174655 | N00312208 | N00079274 | N00694382 |
|---|---|---|---|---|---|
| Supervised | 0.5854 | 0.6484 | 0.7536 | 0.7087 | 0.6479 |
| Transfer | 0.6130 | 0.6909 | 0.7658 | 0.7163 | 0.6752 |
| Zero-shot | 0.5990 | 0.6752 | 0.7576 | 0.7036 | 0.6740 |

### 3.4 Q4. Zero-shot learning

Although there are numerous papers on zero-shot learning (ZSL) in CV and NLP [33, 34, 35], we notice that ZSL was hardly mentioned in tabular domain. In this experiment, we refer to the ZSL scenario mentioned by S(4) of Fig. 1 where we split the raw table into three equal-size subsets. Three subsets have distinct columns. For the *zero-shot* setting, the model learns from set1+set2 and is tested on set3 without further training. In this scenario, the model needs to leverage the learned knowledge from set1 and set2 to support the inference on a new table set3. Besides, we design two baselines for comparison: *supervised* where the model learns from set3 and predicts on set3 and *transfer* where the model learns from set1+set2 and continues to be finetuned on set3. Results are in Table 5. We surprisingly find the ZSL model gets better performance than the supervised one on average. It boils down to that (1) ZSL TransTab succeeds to retain the learned knowledge from set1+set2 for predicting on a new table (set3) and (2) ZSL can benefit from more data (set1+set2) than the supervised (set3 only). Meanwhile, the transfer model takes the advantage of set1+set2 and is adapted for set3 by finetuning, hence reaches the best performance. Similarly, we witness that TransTab is able to make zero-shot predictions on public datasets as in Table 11.

Additional sensitivity check is provided by Fig. 6 where we vary the overlap ratio of two subsets from the same dataset. We witness that our model makes reasonable predictions even if the training set has no column overlap with the test set.

### 3.5 Q5. Supervised and self-supervised pretraining

We take experiments to compare the proposed VPCL with the vanilla transfer learning strategy, as in Table 6. We observe that the vanilla strategy harms the performance on two datasets while VPCL always brings positive effect for finetuning. Besides, we conduct experiments on varying the number of partitions and show the average AUROC on all five datasets, shown by Fig. 4. We specify that VPCL demonstrates an advantage over self-VPCL when we increase the partition numbers.

We also explore if pretraining works on public datasets. Results in Table 12 somewhat match our expectations that pretraining on unrelated tabular data usually yields few benefits for finetuning because these tables define totally different columns and targeted tasks. We also show the ablation on the number of partitions by Fig. 5 where VPCL consistently outperforms the *Supervised* baseline.

Table 6: Test AUROC on clinical trial datasets under the **across-table pretraining plus finetuning** setting. *Supervised*: baseline supervised model; *Transfer*: vanilla supervised transfer learning. Red shows the one worse than the *Supervised* baseline.

| TransTab | N00041119 | N00174655 | N00312208 | N00079274 | N00694382 |
|---|---|---|---|---|---|
| Supervised | 0.6313 | 0.8348 | 0.7444 | 0.6885 | 0.7293 |
| Transfer | **0.6424** | 0.8183 | 0.7458 | 0.6928 | 0.7239 |
| Self-VPCL | 0.6412 | 0.8577 | 0.7486 | **0.7069** | 0.7348 |
| VPCL | 0.6405 | **0.8583** | **0.7517** | 0.7063 | **0.7392** |

Nevertheless, it is still worth investigating the table phenotypes to aggregate tables which are more likely to benefit from each other by transfer learning.

## 4 Related Works

**Tabular Prediction.** To enhance tabular predictions, numerous recent works try to design new algorithms [28, 36, 37, 30, 38, 32, 5, 10, 7, 39, 40, 41, 42, 43, 44, 45]. However, it was argued that boosting algorithms and MLPs are still the competitive choices for tabular data modeling, especially when the sample size is small [32, 46, 39, 47]. To alleviate label scarcity issue, SSL pretraining on unlabeled tabular data was introduced [2, 24, 10, 9, 11]. Nonetheless, none of them is *transferable* across tables then is able to extend the success of pretraining to the tabular domain. For practical tabular predictions, the common case is that we own a lot of labeled samples collected with diverse protocols hence heavy preprocessing is needed to align them by either dropping many samples or many features. By contrast, TransTab accepts variable-column tables and therefore can learn from different tables at scale and transfer to the target task. Also, it can support diverse tabular prediction tasks as depicted in Fig. 1, which cannot be done by off-the-shelf tabular methods.

**Transfer learning.** Transfer learning (TL) has long been a popular research field since the proposal of ImageNet [48], which gives rise to splendid works on utilizing supervised pretraining on a large general database and finetune on a small downstream task [49, 50, 51, 52, 53]. TL is also fast-growing in NLP beginning at BERT [20], which often leverages web-scale unlabeled texts for self-supervised pretraining and then applies to specific tasks [34, 54, 55, 56, 57]. However, few work was on TL in tabular predictions. As mentioned in §1, TransTab paves the way for effective tabular TL by establishing a feature processing protocol that applies for most table inputs, such that it shares knowledge across tables.

**Self-supervised learning & contrastive learning.** SSL uses unlabeled data with pretext tasks to learn useful representations and most of them are in CV and NLP [20, 17, 15, 16, 58, 23, 59, 60, 61, 62, 63]. Recent SSL tabular models can be classified into *reconstruction* and *contrastive* based methods: TabNet [30] and VIME [2] try to recover the corrupted inputs with auto-encoding loss; SCARF [11] takes a SimCLR-like [64] contrastive loss between the sample and its corrupted version; SubTab [9] takes a combination of both. Nevertheless, all fail to learn transferable models across tables such that cannot benefit from pretraining with scale. Contrastive learning can also be applied to supervised learning by leveraging class labels to build positive samples [26]. Our work extends it to to the tabular domain, which we prove works better than vanilla supervised pretraining. The vertical partition sampling also enjoys high query speed from large databases which are often column-oriented [25]. Another line of research takes table pretraining table semantic parsing [65, 66, 67, 68, 69] or table-to-text generation [70, 71]. But these methods either encode the whole table instead of each row or do not demonstrate to benefit tabular prediction yet.

## 5 Conclusion

This paper proposes TransTab that accepts variable-column inputs. By the proposed vertical partition contrastive learning, it can benefit from supervised pretraining from multiple tabular datasets with low memory cost. We envision it to be the basis of tabular foundation models and widely used to tabular-related applications in the future.

## Acknowledgement

This work was supported by NSF award SCH-2205289, SCH-2014438, IIS-1838042, NIH award R01 1R01NS107291-01.

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
