# OpenReview forum: "TransTab: Learning Transferable Tabular Transformers Across Tables"
_NeurIPS.cc/2022/Conference — NeurIPS 2022 Accept_

### Official Review · Reviewer_THBY · 2022-07-01

**Rating:** 7
**Confidence:** 4
**Soundness:** 3 good
**Presentation:** 3 good
**Contribution:** 3 good

**Summary:**

This paper focuses on the transferability of tabular data classification methods. It proposes three novel settings to evaluate the model transferability in terms of columns: column overlapping, column increment, and zero-shot. It also proposed a novel method combining self-supervised and supervised pre-training.

**Questions:**

* line 98: Why not assign an embedding vector when $x_b \neq 1$? Do you compare them?
* line 105: There is no value limit for $x_u$. Do you encounter any problem without scaling $E_{u}$?
* line 214: (5) -> (2) in Fig1?

**Limitations:**

* It would be meaningful to compare the methods focusing on numerical tables with those focusing on text tables.

**Strengths And Weaknesses:**

### Strength
* Three novel settings to evaluate the model transferability on tabular data classification. Transferability is an important research topic.
* A novel method based on (self-)supervised pre-training for tabular data classification which is more accurate and transferable.

### Weakness
* Incorrect claim in line 109: $E$ is not contextualized. To get contextualized embedding, the input embeddings should interact with each other, but not simply concatenization.
* Feature incremental learning setting is unclear.
* No baseline results (e.g. VIME and SCARF) for zero-shot setting.
* This paper assumes tables are matrix-like and column types are given, which hiders its transferability. Many papers in the NLP community have explored to process tables under a more flexible setting:
  * Wang et al. [Robust (Controlled) Table-to-Text Generation with Structure-Aware Equivariance Learning](https://arxiv.org/pdf/2205.03972). NAACL 2022
  * Yang et al. [TableFormer: Robust Transformer Modeling for Table-Text Encoding](https://arxiv.org/pdf/2203.00274.pdf). ACL 2022
  * Wang et al. [Retrieving complex tables with multi-granular graph representation learning](https://arxiv.org/pdf/2105.01736). SIGIR 2021

---

> ### Author Response · Authors · 2022-07-31
> **Response to Reviewer THBY**
>
> We thank Reviewer THBY for the  helpful feedback. Besides our general response above, please see our specific response below.
>
> **Line 109 claim**
>
> > Incorrect claim in line 109: E is not contextualized.
>
> It is correct that $E$ is not contextualized in terms of feature level when applying interactions like attention. However, *TransTab* treats each feature in two parts: column name and cell value, e.g., value *20* under the column *age*. Here, when we concatenate *20* with *age*, the value is contextualized. Thanks for the suggestion and we rephrase this part in the new version.
>
>
>
> **Feature incremental learning setting**
>
> > Feature incremental learning setting is unclear.
>
> We clarified the settings in the new version. Please refer to the response to **Reviewer teon**: **Feature incremental learning v.s. transfer learning**, **Line 214-215**, **Experiments regarding feature incremental learning**, and **Establishment of subsets**. We have updated the experiment descriptions in the new version.
>
>
>
> **Baselines in zeroshot learning**
>
> > No baseline results (e.g. VIME and SCARF) for zero-shot setting.
>
> Although VIME and SCARF are all self-supervised methods, they are unable to deal with zero-shot prediction because they are trained either with reconstruction or with the contrastive objective. After pretraining, these models need an additional classification head and further supervised fine-tuning. These methods do not handle the case when we only have limited  labeled samples because pretraining and supervised learning are infeasible. Moreover, they need to be retrained every time when table structure varies.
>
>
>
> **Related papers in NLP**
>
> > This paper assumes tables are matrix-like and column types are given, which hiders its transferability. Many papers in the NLP community have explored to process tables under a more flexible setting:
>
> Thank  for providing the literature on table retrieval and table-to-text generation. We will add them to the related works. Nonetheless, these works are on quite different tasks from ours. And it is unclear if they contribute to superior tabular prediction performances. Specifically, both table-text generation and table retrieval methods encode the whole table to an embedding while *TransTab* encodes each row of tables. We agree it is interesting future work to extend *TransTab* to more settings, e.g., nested table.
>
>
>
> **Line 98 binary feature**
>
> > line 98: Why not assign an embedding vector when $x_b \neq 1$? Do you compare them?
>
>  We did consider the implementation that adds an indicator embedding for True or False in binary features. We compared it with the current method and found that did not achieve better results. Moreover, our current method significantly reduces computational cost for high-dimensional sparse tables because all False binary columns are not included in encoding.
>
>
>
>
> **Line 105 value limit**
>
> > line 105: There is no value limit for $x_u$. Do you encounter any problem without scaling $E_u$?
>
> In practice, we scale $x_u$ by standardization or normalization such that $E_u$ is numerically stable. We added this detail in our new version on Line 105.
>
>
>
> **Line 214 typo**
>
> >line 214: (5) -> (2) in Fig1?
>
> Thanks, we fixed this typo in the new version.
>
>
>
> **More baselines**
>
> > It would be meaningful to compare the methods focusing on numerical tables with those focusing on text tables.
>
> This is an interesting future work, which probably needs additional methods to handle different data types and their semantic relations.
> We could draw ideas from the literature on table generation, table retrieval, and table semantic parsing to tackle this new setting. But it is probably beyond the scope of this paper.

---

### Official Review · Reviewer_teon · 2022-07-11

**Rating:** 7
**Confidence:** 4
**Soundness:** 2 fair
**Presentation:** 3 good
**Contribution:** 3 good

**Summary:**

This paper presents a tabular learning framework that covers transfer learning across tables, zero-shot inference, feature incremental learning, pre-training, and finetuning. This approach does not assume that columns in the table are fixed and work even with variable column tables. The authors propose two Contrastive Learning-based pre-training approaches by vertically partitioning the tables. This pre-training approach is feasible since the columns can vary across tables, making self-supervised and supervised pre-training possible. The transformer model proposed performs significantly better in all the claimed settings (transfer learning across tables, zero-shot inference, feature incremental learning, pre-training, and fine-tuning). In addition, the authors also introduce *clinical trial mortality prediction* tabular dataset.

**Questions:**

1. How the partitions of set 1,2,3 are created in transfer learning, zero-shot inference, and feature incremental learning settings?
2. Are these sets created by randomly partitioning for every seed? Or the partition is fixed for all seeds?
3. Are subsets in feature incremental learning and zero-shot inference settings the same?

**Strengths And Weaknesses:**

**Pros**
* The proposed contrastive learning methods are computationally cheaper.
* This variable column approach is really useful when the tables have too many columns and encoding them will be difficult in current existing transformers for tabular data (e.g. TaBERT).

**Cons**
* Setting for *Feature incremental learning* and *Transfer learning* seems very similar. (Dividing the dataset into three sets containing an equal number of columns and first training on set 1&2 then train on set 3 vs the transfer learning setting in the paper)
* line 214-215 is confusing (incomplete)
* In Feature incremental learning, no comparisons on how the performance on set1 after training on set1+set2; set1, set2, set1+set2 after training on set1+set2+set3. Will the performance on previous sets decrease? If yes, How to mitigate that?

---

> ### Author Response · Authors · 2022-07-31
> **Response to Reviewer teon**
>
> We thank Reviewer teon for the helpful feedback. Besides our general response at the beginning, we provided additional details to address some specific comments.
>
>
>
> **Feature incremental learning v.s. transfer learning**
>
> > Setting for Feature incremental learning and Transfer learning seems very similar.
>
> It is true that both applications are conceptually similar.  However, they are still different:
>
> - For *feature incremental learning* when we have set1, set2, and set3 which built incrementally (set2 includes all columns in set1, set3 include all in set2), the object is to involve all three in the same round of training (w/ supervised loss) and enhance the prediction for set3.
> - For *transfer learning* when we have three equal-sized subsets (columns are assigned randomly into the three sets), the training process has two stages: pretraining + finetuning. In the first stage, the model is trained on set1+set2 using contrastive pretraining, namely VPCL in our paper (supervised or self-supervised); in the second, the model is trained on set3 only (supervised).
>
> We discriminate between these two settings because they apply to different scenarios. Transfer learning based on VPCL is for learning from a wide range of data to build a foundation model good for adapting to downstream tasks; Feature incremental learning is for making the best of all data from the same domain.
>
>
>
> **Line 214-215**
>
> > line 214-215 is confusing (incomplete)
>
> We rephrased and extended that part on the new version on Line 213-216. We split the raw dataset into three subsets: set1, 2, and 3. Baseline methods apply to two scenarios: (1) learning from all data that only have features of set1 and (2) learning from data from set3 only. We report the best of the two. *Transtab* applies to learning from all three subsets.
>
>
>
> **Experiments regarding feature incremental learning**
>
> > In Feature incremental learning, no comparisons on how the performance on set1 after training on set1+set2; set1, set2, set1+set2 after training on set1+set2+set3.
>
> It is feasible to add those additional experiments, but the primary object of feature incremental learning is to enhance the performance on set3. We do not train *TransTab* stepwise on set1, 2, and then 3. Instead, all three sets are used in one training round simultaneously. The reviewer's proposal fits the target of transfer learning: improving performance on each dataset by learning across datasets. We refer the reviewer to Table 4 in the paper which illustrates the results of transfer learning and Table 6 that shows the average improvement led by contrastive pretraining on all datasets.
>
>
>
> **Establishment of subsets**
>
> > How the partitions of set 1,2,3 are created in transfer learning, zero-shot inference, and feature incremental learning settings? Are these sets created by randomly partitioning for every seed? Or the partition is fixed for all seeds? Are subsets in feature incremental learning and zero-shot inference settings the same?
>
>
> For experiments in Sec. 3.2, 3.3, 3.4, we create subsets randomly with a fixed seed, respectively. That is, the subsets vary across these sections.
>
> - Feature incremental learning. The columns are split into three distinct parts ${v_1,v_2,v_3}$. Set1 contains $v_1$, set2 contains $v_1,v_2$, and set3 has $v_1,v_2,v_3$. Three sets have an equal number of samples.
> - Transfer learning. The columns are split into two parts $v_1,v_2$ where $v_1$ and $v_2$ have 50% of elements overlapped. Two sets have an equal number of samples.
> - Zeroshot learning. The columns are split into three distinct parts ${v_1,v_2,v_3}$. Set1 contains $v_1$, set2 contains $v_2$, set3 contains $v_3$. Three sets have an equal number of samples.
>
> We add this explanation to appendix D of the new version.

---

> > ### Comment · Reviewer_teon · 2022-08-09
> > **Thanks for the response.**
> >
> > Most of my queries/doubts/concerns are answered and I also increased the rating to 7.

---

### Official Review · Reviewer_MTvW · 2022-07-12

**Rating:** 7
**Confidence:** 4
**Soundness:** 3 good
**Presentation:** 3 good
**Contribution:** 3 good

**Summary:**

This paper proposed to relax fixed table structures by introducing a Transferable Tabular Transformer (TransTab) for tables. They basically convert each row into a feature vector and then apply stacked transformers for feature encoding. There are several advantages of this encoding: (1) it can deal with the tables that have different number of columns; (2) it is easier to transfer the knowledge learned from different columns. They conduct experiments on one clinical dataset and several public datasets under four different settings: supervised learning, feature incremental learning, transfer learning, and zero-shot learning. The empirical results show that the proposed approach outperform the baselines in the literature. They also showed that in the zero-shot learning scenario, they can almost match the performance of pretraining plus fine-tuning.

**Questions:**

see strengths and weaknesses for details

**Limitations:**

This paper has sufficiently addressed the limitations.

**Strengths And Weaknesses:**

- Originality: The main idea in this paper is a good combination of several ideas proposed in the literature. With modifications and adaptations, it worked and yield promising results on several datasets.

- Quality: The proposed approach is technically sound and the experimental results showed that it outperformed several strong baselines for tabular data prediction. Although the results are impressive, I have several comments:
  - One of the main advantages advertised in the paper is that the proposed method could easily extend to feature incremental learning, pretraining+finetuning, and zero-shot inference. In the paper "existing works only cover vanilla supervised learning and fixed-table pretraining due to the fixed-column assumption." In my point of view, this is overclaiming. Not all existing works only cover vanilla supervised learning. For example, those transformer-based architectures like TabTrans, FT-Trans, can be easily adapted to those settings.
  - The zero-shot performance in Table 5 seems surprising to me. How do you split the table into three distinct sets? Do you do random split and how many random seeds have you tried? I would imagine a split that during the training, the model mostly sees Categorical and Binary features while during test it mainly sees Numerical features. In this way, I don't think the model is able to do zero-shot transfer. Moreover, can you try a setting that you manually control the number of categorical, binary, and numerical feature in both training and testing and see how does the model generalize?
  - In addition to the quantitative results, I would also like to see some qualitative analysis of the transferability of the model. What does the data look like and why the model is able to do the transfer?
  - How does the self-supervised pretraining affect the performance? I would like to see an ablation study where you only training the model with the direct supervision signals. This could help understand how much of the improvement is from the architecture design and how much improvement is from the self-supervised pretraining.

- Clarity: In general, this paper is well-organized and easy to follow. I think section 2.4 could be better explained: what's the definition of $v_i^k$ in line 133? How do you compute $\psi$ in equation 4?

- Significance: This paper achieved strong results across a range of different datasets. Although the experiments are not comprehensive enough for the readers to understand every aspect of the system, I think it still sets a strong baseline and a good reference for the future work in this direction.

---

> ### Author Response · Authors · 2022-07-31
> **Response to Reviewer *MTvW* (1/2)**
>
> We thank Reviewer MTvW for the helpful feedback to our work. We addressed most of Reviewer MTvW's comments in our general response above. And we will provide additional responses to specific comments below.
>
>
>
> **Justification of phrases**
>
> > In the paper "existing works only cover vanilla supervised learning and fixed-table pretraining due to the fixed-column assumption." In my point of view, this is overclaiming. Not all existing works only cover vanilla supervised learning.
>
> We agree that our original sentence may cause misunderstanding. We intend to illustrate that due to the fixed-column assumption, most existing works only handle supervised learning or pretraining on the same-structure tables. We rephrased that in the new version.
>
>
>
> **Zeroshot prediction**
>
> > The zero-shot performance in Table 5 seems surprising to me. How do you split the table into three distinct sets? Do you do random split and how many random seeds have you tried? ... Moreover, can you try a setting that you manually control the number of categorical, binary, and numerical feature in both training and testing and see how does the model generalize?
>
> In the zero-shot learning experiments, we split the table columns into three equal-sized subsets where columns are 50\% mutually overlapped (e.g., set1 and set2 have 50\% same columns, set2 and set3 have 50\% same columns). Likewise, each subset has 1/3 number of samples. We follow the setting specified in Table 2 caption. All the results are averaged over 10 runs with different random seeds. During this process, the subset columns are fixed but train/test splits are changed. We add more experiments to dive deep into the zero-shot prediction capability of our method. To be specific, we split the dataset into two subsets with no sample overlaps and test zero-shot performance with column overlap ratio varying from 0 (non-overlap) to 1.
> The corresponding figure is added in the new version (Fig. 6 in Appendix). Nonetheless, we agree with ZSL that the problem is very challenging when the training tables and testing tables are highly mismatched.
>
>
> | AUC\overlap ratio | 0      | 0.2    | 0.5    | 0.8    | 1.0    |
> | ----------------- | ------ | ------ | ------ | ------ | ------ |
> | credit-g          | 0.5584 | 0.5740 | 0.6241 | 0.6441 | 0.7612 |
> | credit-a          | 0.8118 | 0.8128 | 0.8583 | 0.8621 | 0.8697 |
> | dress-s           | 0.5640 | 0.5663 | 0.5847 | 0.5740 | 0.7011 |
> | cylinder-b        | 0.5279 | 0.5461 | 0.6657 | 0.6509 | 0.6550 |
>
>
>
> **Qualitative analysis of the transferability**
>
> > In addition to the quantitative results, I would also like to see some qualitative analysis of the transferability of the model.
>
> We added some case studies and discussed why transfer learning and zero-shot prediction across them are feasible.  Two clinical trial datasets are illustrated below. We observe several shared columns across the two datasets. Moreover, there are columns named differently but sharing similar meanings, e.g., "adverse effect: infection" and "adverse effect: infection without neutropenia(specify)". *TransTab* can be effective in both scenarios.
>
> |      | adverse effect: nausea | adverse effect: vomiting | adverse effect: asthenia | adverse effect: infection |
> | ---- | ---------------------- | ------------------------ | ------------------------ | ------------------------- |
> | 0    | 0                      | 0                        | 0                        | 0                         |
> | 1    | 0                      | 0                        | 0                        | 0                         |
> | 2    | 0                      | 0                        | 0                        | 0                         |
>
> |      | adverse effect: febrile neutropenia | adverse effect: infection (documented clinically) | adverse effect: infection without neutropenia(specify) |
> | ---- | ----------------------------------: | ------------------------------------------------: | -----------------------------------------------------: |
> | 0    |                                   0 |                                                 0 |                                                      0 |
> | 1    |                                   0 |                                                 0 |                                                      0 |
> | 2    |                                   0 |                                                 0 |                                                      0 |

---

> > ### Author Response · Authors · 2022-07-31
> > **Response to Reviewer *MTvW* (2/2)**
> >
> > **Ablation on self-supervised pretraining**
> >
> > > How does the self-supervised pretraining affect the performance? I would like to see an ablation study where you only training the model with the direct supervision signals.
> >
> > In Section 3.1, our method uses the direct supervision signals without a pretraining step, which might be the requested ablation study. We further discussed the benefit of pretraining in Section 3.5. In particular, the proposed VPCL pretraining generally leads to better performance on clinical trial mortality prediction datasets. We clarified these settings in the experiment section, on Line 203.
> >
> >
> >
> > **Clarity**
> >
> > > I think section 2.4 could be better explained: what's the definition of vik in line 133? How do you compute ψ in equation 4?
> >
> > As defined in Line 133, $v$ are vertical partitions of the table, e.g., $v_1$ are rows under column 1, $v_2$ are rows under column2, and so on. In Line 137, $\psi$ is the cosine similarity of two vectors. We improved Sec 2.4 to address these writing issues in the new version.

---

> > > ### Comment · Reviewer_MTvW · 2022-08-09
> > > **Thanks for the detailed responses**
> > >
> > > Thanks for addressing my questions and concerns and I am happy to raise my score to 7.

---

### Author Response · Authors · 2022-07-31
**General response to all reviewers**

We thank the reviewers for their thoughtful and constructive reviews. Most comments center around the settings and experiments of **feature incremental learning** and **zero-shot learning**. In response, we have updated our manuscript to include a detailed explanation of the experiment settings. We also ran additional experiments to test the sensitivity of zero-shot learning w.r.t. the overlapping ratio of table columns.

Here is the change summary:

- Sec. 2.2, L105: added a footnote to explain how to avoid the numerical issue when encoding numerical features.
- Sec 2.2, L109: rephrased the concept w.r.t. the cell embedding contextualization to avoid misunderstandings.
- Sec 2.4, L133: further explained $v_i^k$ in $x_i$.
- Sec 2.4, L137: further explained the similarity function $\psi$.
- Sec 3.1, L203: further explained the supervised learning setting.
- Sec 3.2, L213-L216: rephrased the experiment settings of feature incremental learning.
- Sec 3.4, L242-L244: added an experiment to check the sensitivity of zero-shot learning w.r.t. the overlapping ratio of columns of two tables.
- Sec 4, L287: added related works suggested by Reviewer THBY and discussed the difference to our method.
- Appendix D, L592: added an introduction on how to build the subsets for three experiments (Sec 3.2, 3.3, 3.4).
- Appendix, Fig.6: added figures for the sensitivity evaluation of zero-shot prediction.

---

### Public Comment · ~Talip_Ucar2 · 2023-01-01
**The paper's similarity to SubTab**

Thank you for your work. I went through the paper since the topics around representation learning, transfer learning and incremental feature learning in the context of tabular data are in my area of interests. As a summary of what is described below, TransTab claims some of the contributions made by SubTab and there is a mischaracterisation and a lack of proper citation regarding our work. In details:

About self-supervised learning, the idea of "Vertical-Partition Contrastive Learning (VPCL)" seems to be very close (if not same) to my previous NeurIPS paper, SubTab [1]. Authors passingly cited SubTab together with other self-supervised methods, but not in the context of VPCL. Specifically, SubTab proposed partitioning features of tabular data for contrastive learning and is the first such model to do so. Considering that, would you please elaborate on what is novel in TransTab? If not, I believe that you should at least cite SubTab and discuss it in your paper, especially since the number of such papers are limited in the tabular domain.

Apart from that, SubTab would be the closest baseline, if not the SOTA, in the experiments listed in this paper, and it is interesting to see that the authors did not include it as one of the benchmarks. Would you please elaborate on how this model performs compared to SubTab?

Moreover, SubTab is a flexible framework that opens up opportunities for other tasks such as:

    - 1) Transfer learning via shared features across many tabular datasets (a common occurrence in tabular setting)

    - 2) Incremental feature learning as one acquires new features over time

As a natural extension of SubTab, we discussed how to implement these ideas in Figure A6 on the Page-18 (Section D.2) of the Appendix in the paper. These ideas were also presented in NeurIPS's presentation and poster sessions in 2021.

     - Paper: https://arxiv.org/pdf/2110.04361.pdf

     - Presentation (Slide 47 onwards): https://github.com/AstraZeneca/SubTab/blob/main/assets/NeurIPS_2021_slides.pdf

TransTab's experiments have similar setups to those we discussed in the paper and presentation. It is nice to see that we could see them as follow-up on your work. It could have been nice if the authors cited our work in this regard as well.

Thank you for your work again, and I am looking forward to the authors' comments.

### Reference:

[1]  Talip Ucar, Ehsan Hajiramezanali, and Lindsay Edwards. SubTab: Subsetting features of tabular data for self-supervised representation learning. Advances in Neural Information Processing Systems, 34, 2021.

---

> ### Public Comment · ~Zifeng_Wang3 · 2023-01-01
> **thanks for your interest**
>
> Hi, Talip, thanks for your interest in our work. I will try to clarify your concerns.
>
>
> 1. Difference. Generating two views from the same row for contrastive learning on tabular data was mentioned earlier in 2020 [1]. The basic idea of SubTab is similar: using contrastive learning by generating two views from the same row to build positive samples while treating all the others from different rows as negative. Subtab differs in the corruption strategy and it builds augmented views by deletion. This setup is closest to our baseline setup named "Self-supervised VPCL" but our method considers modeling the column semantics besides the cell values. By contrast, our main method VPCL conducts a "supervised" contrastive learning paradigm: it treats partitions from rows with the same class as the positive, which we believe was not mentioned in previous relevant methods.
>
>
> 2. Baseline. We thank you for noticing us about subtab. We compared with SCARF [2] which also follows the tabular contrastive learning that builds positive samples through corruptions. We agree it would be interesting to also compare with subtab. Moreover, it might be more appealing to explore the possibility to make the benefit from both subtab and transtab: with explicitly modeling column semantics and better contrastive learning to obtain better pretrained tabular models.
>
>
> 3. Motivation. The subtab paper mentioned the potential application to transfer learning and zero-shot setups in the discussion section but we did not find empirical evaluation from the original paper. We also believe the principle of TransTab on fulfilling transferability differs substantially from subtab: transtab explicitly takes column semantics into account for tabular encoding, which endows transfer knowledge across datasets by reusing the learned semantics, but subtab extracts cell values only and (maybe) achieves transfer learning in another route.
>
> [1] Yao, T., Yi, X., Cheng, D. Z., Yu, F., Chen, T., Menon, A., ... & Ettinger, E. (2020). Self-supervised Learning for Large-scale Item Recommendations. arXiv preprint arXiv:2007.12865.
>
> [2] Bahri, D., Jiang, H., Tay, Y., & Metzler, D. (2021, September). Scarf: Self-Supervised Contrastive Learning using Random Feature Corruption. In International Conference on Learning Representations.

---

> > ### Public Comment · ~Talip_Ucar2 · 2023-01-02
> > **Follow-up**
> >
> > Thanks, Zifeng, for your prompt response.
> >
> > Regarding VPCL, my point is on self-supervised part of your paper, specifically partitioning features to multiple sets for self-supervised learning. This is already proposed in the SubTab. I think that SubTab should be cited for this. I agree that TransTab extend it to supervised  domain by using labels.  I believe that once SubTab is cited, you can make a further distinction between SubTab and TransTab as you did in your response. This will give a clear vision of how things have evolved in the past few years.
> >
> >
> > Regarding the works [1, 2] that you cited:
> >
> > In [1], there is no partitioning of features to multiple sets. They propose different masking strategies called Random Feature Masking (RFM)  and Correlated Feature Masking (CFM). And they use the full feature set to generate two masked version of the same data as shown in Figure-3 of the paper.
> >
> > In [2], SCARF is an adaptation of SimCLR and again uses the full feature set. They generate two augmented version of the same data for contrastive learning. SCARF can be considered just an extension of SubTab without feature partitioning and decoder. We talk about this setup in Section-E of Appendix in SubTab since we initially experimented with this setup. In our experiments with SimCLR-like models, SubTab has always performed better. Thus, I believe that SubTab is a stronger baseline than SCARF.
> >
> > The works [1, 2] and many similar ones differ only in how they augment the input data. They all use the full set of features. These approaches are very similar to De-noising Autoencoder, in which they replace reconstruction loss with contrastive loss. They mainly differ from one another in masking and noising strategies. These strategies would not be able to extend to transfer learning or incremental feature learning easily in the context of tabular data.
> >
> > SubTab is the first work that proposed partitioning features to multiple-sets for self-supervised learning. Our main motivation for partitioning features was to do transfer learning (by taking advantage of shared features across different datasets) and incremental feature learning in the future as we discussed in the paper as a future work. We also introduced the nomenclature that includes terms such as "view" to refer to each partition, and "overlap" to refer to shared features between partitions, both of which seem to be adapted by TransTab as well.
> >
> > As a last concern, in TransTab, SubTab is put in the category of models that "assume the table structure keeps fixed in training and testing". As we show in the experiments listed in Figure-5 of SubTab, this is clearly not true. So, I really appreciate it if you can revise your paper in this regard.
> >
> > Kind regards,
> >
> >
> > ### Reference:
> >
> > [1] Yao, T., Yi, X., Cheng, D. Z., Yu, F., Chen, T., Menon, A., ... & Ettinger, E. (2020). Self-supervised Learning for Large-scale Item Recommendations. arXiv preprint arXiv:2007.12865.
> >
> > [2] Bahri, D., Jiang, H., Tay, Y., & Metzler, D. (2021, September). Scarf: Self-Supervised Contrastive Learning using Random Feature Corruption. In International Conference on Learning Representations.
> >
> > [3] Talip Ucar, Ehsan Hajiramezanali, and Lindsay Edwards. SubTab: Subsetting features of tabular data for self-supervised representation learning. Advances in Neural Information Processing Systems, 34, 2021.

---

### Meta-Review · Area_Chair_jeiJ · 2022-08-22

**Recommendation:** Accept
**Confidence:** Certain

**Metareview:**

This work introduces and evaluates a general scheme to feature-ize tabular data, and methods for (self-supervised) pre-training over the same, with a focus is on learning transferable representations.

Reviewers were unanimous that the approach proposed constitutes a flexible, practical approach that borrows and brings together existing SOTA techniques. Some questions about the specific settings concerned in the evaluation (and distinctions between them) were sufficiently addressed during the response period. Empirical results show consistent gains over the baselines on the tasks considered.

An additional suggestion: one might naively anticipate that transfer learning for tables is not particularly promising given the very different semantics two arbitrary tables might have. However, the scenarios considered here involve settings in which transfer seems a priori reasonable; I might suggest the authors address this upfront, and explicitly outline the conditions under which transfer learning for tables is anticipated to work (and what assumptions are necessary for such cases), and where it is not.

**Award:**

No

---

### Decision · Program_Chairs · 2022-09-14

Accept